# Nano-Sized Calcium Copper Titanate for the Fabrication of High Dielectric Constant Functional Ceramic–Polymer Composites

**DOI:** 10.3390/polym14204328

**Published:** 2022-10-14

**Authors:** Jinfa Ou, Yonghui Chen, Jiafu Zhao, Shaojuan Luo, Ka Wai Wong, Ka Ming Ng

**Affiliations:** 1School of Chemical Engineering and Light Industry, Guangdong University of Technology, Guangzhou 510006, China; 2International Collaborative Laboratory of 2D Materials for Optoelectronic Science and Technology, Shenzhen University, Shenzhen 518060, China; 3Department of Chemical and Biomolecular Engineering, The Hong Kong University of Science and Technology, Clear Water Bay, Kowloon 999077, Hong Kong

**Keywords:** CaCu_3_Ti_4_O_12_, Cu-deficiency, PVDF matrix, dielectric property, post-treatment

## Abstract

A novel calcium copper titanate (CaCu_3_Ti_4_O_12_)–polyvinylidene fluoride composite (CCTO@PVDF) with Cu-deficiency was successfully prepared through the molten salt-assisted method. The morphology and structure of polymer composites uniformly incorporated with CCTO nanocrystals were characterized. At the same volume fraction, the CCTOs with Cu-deficiency displayed higher dielectric constants than those without post-treatment. A relatively high dielectric constant of 939 was obtained at 64% vol% CCTO@PVDF content, 78 times that of pure PVDF. The high dielectric constants of these composites were attributed to the homogeneous dispersion and interfacial polarization of the CCTO into the PVDF matrix. These composites also have prospective applications in high-frequency regions (10^6^ Hz). The enhancement of the dielectric constant was predicted in several theoretical models, among which the EMT and Yamada models agreed well with the experimental results, indicating the excellent distribution in the polymer matrix.

## 1. Introduction

Promoting new electronic energy relies on supercapacitors, sensors, embedded capacitors, and mobile electronic storage devices [1,2,3,4]. Dielectric materials play a significant role in capacitors and electronic devices, which generally require easy processing, high energy density, and excellent flexibility [5,6]. Universally, polymers feature good flexibility, high electron breakdown strength, and environmental tolerance, but very low dielectric properties (less than 5), which is far below the practical application value of 50. To replenish this shortage, the filling of high-*k* materials can significantly improve the dielectric properties of polymers, especially the ceramic particles, including titanium dioxide (TiO_2_), barium titanate (BT), barium strontium titanate (BST), and lead zirconate titanate (PZT) [7,8,9,10,11,12,13,14]. Thus, ceramic-based polymer materials have attracted significant interest from the scientific and engineering community for their potential in charge storage applications.

For polymer dielectric capacitors, the energy density *U*_e_ is proportional to its dielectric constant (electrical permittivity) *K* and the square of the electric field *E*: [9]
(1)$$U_{e}=\frac{1}{2}K\varepsilon_{0}E^{2}$$
where *ɛ*_0_ is the vacuum dielectric constant, and the improved performance of the dielectric materials is critical to the enhanced performance of the polymer dielectric capacitors and their applications. Up to date, many studies have reported the preparation of polymer matrix nanocomposites with high permittivity by filling high-dielectric ceramic materials, such as BaTiO_3_/epoxy composites (40 vol%), BaTiO_3_/polyethersulfone (PES) composites (40 vol%), and CaCu_3_Ti_4_O_12_ (CCTO)/polyethersulfone composites (40 vol%) [15,16,17,18,19,20]. Among them, CCTO ceramic possesses a giant dielectric constant and remains unchanged in a wide temperature range (100–500 K). Chao et al. reported CCTO/PVDF nanocomposites (50 vol%) with a high dielectric property of about 62.3 and a low loss of 0.081 using a series of synthesis methods [21]. Hu et al. studied a unique polymer composite fabricated with CCTO nanofibers (20 vol%) and PVDF via the electrospinning method. Liu’s research revealed that CCTO nanowire/epoxy efficiently improved the dielectric permittivity and mechanical properties of ceramic-based polymer composites [22,23]. Wang et al. reported that the dielectric constant of CCTO/PVDF composite materials with 50% CCTO achieved a maximum value of almost 50, which is five times higher than the pure PVDF. Moreover, a series of studies revealed that the inorganic conductive fillers with relatively high conductivity may lead to a high dielectric constant of ceramic/polymer materials, such as Ba_0.6_Sr_0.4_TiO_3_/Poly (vinylidene fluoride) composites, Ag-NBCTO-PVDF composites, and FeAlSi/BaTiO_3_/epoxy composites [24,25,26,27].

Herein, this work presented a novel molten salt-assisted method to synthesize CCTO nanocrystals with a Cu-deficiency structure. The CCTO fillers were uniformly dispersed in the PVDF polymer, and the dielectric permittivity of the Cu-deficiency CCTO@PVDF was dramatically increased. The relatively high dielectric constant of the nanocomposite (64 vol%) is almost 78 times that of the neat PVDF matrix. On the other hand, through the simulation and prediction of various theoretical models, the dielectric constant and structure under the EMT and Yamada models are highly similar to the experimental results. They show that the post-treatment of CCTO can effectively improve the dispersion and dielectric properties of ceramic polymer composites.

## 2. Materials and Methods

### 2.1. Chemicals

Titanium isopropoxide (Ti[OCH(CH_3_)_2_]_4_, 97%), calcium nitrate tetrahydrate (Ca(NO_3_)_2_·4H_2_O, 99%), copper nitrate hemi(pentahydrate) (Cu(NO_3_)_2_·2.5H_2_O, 98%), acetylacetone (CH_3_COCH_2_OCH_3_, 99%), citric acid (monohydrate) (C_6_H_8_O_7_·H_2_O, 99%), and sodium chloride (NaCl, 99%) were purchased from Sigma-Aldrich Company; ethylene glycol (OHCH_2_CH_2_OH, 99.5%) and ethanol (EtOH, 99%) were obtained from Fluka; PVDF resin powder and graphene nanosheets were obtained from Kynar and Strem Chemicals, respectively. All the chemicals were used as received without any further purification.

### 2.2. Synthesis of CCTO Nanocrystals

The precursor gel solution was synthesized by modifying the reported protocol: Ref. [28]. A total of 0.02 mol (5.88 g) of titanium isopropoxide and 0.02 mol (2.00 g) of acetylacetone were mixed with 0.2 mol (12.42 g) of ethylene glycol under vigorous magnetic stirring to get a yellow solution. Then, 0.02 mol (4.20 g) of citric acid monohydrate was added to the solution and stirred until all the citric acid was dissolved. After that, 0.005 mol (1.18 g) of calcium nitrate tetrahydrate and 0.15 mol (3.48 g) of copper nitrate hemi(pentahydrate) were introduced into the solution with magnetic stirring. Finally, a stable green solution was obtained. The solution was poured into an alumina crucible and placed into the furnace with a heating rate of 10 °C/min to 350 °C for 15 min. The resultant porous black precursor was collected, grounded into powder (about 4.0 g), and mixed with no or a given quantity of sodium chloride (3 and 10 times in weight ratio) in ethanol using the ball milling machine at 300 rpm for 3 h. Then, the dried pre-mixed precursor was calcinated at 700–900 °C for a different time duration, from 1–5 h. The crude product was washed with deionized water several times to remove the molten salt. The CCTO product was labeled as CCTO-*x*-*y* (*x* represented the temperature and *y* represented the time duration). In addition, the selected CCTO sample was dispersed in ethanol by sonication for 2 h. The sample was vacuum dried at 60 °C overnight after being centrifuged. The resultant loose powder was post-treated at 1050 °C for 1 and 4 h and marked as CCTO-1050-1 h and CCTO-1050-4 h.

### 2.3. Preparation of CCTO@PVDF Composite

The synthesized CCTO and PVDF powder were added into a ball milling jar with agate malls to mix at 300 rpm for 30 min thoroughly. The mixture was then molded together for compression at 200 °C for 30 min at a pressure of 100 MPa to form a composite pellet of 0.5 mm in thickness and 12 mm in diameter.

### 2.4. Characterizations

Thermo gravimetrical analysis (TGA) of the black powder precursor was performed on a TGA (Perkin Elmer, UNIX/TGA7) from room temperature to 900 °C in an air atmosphere with a heating rate of 10 °C/min. Phase identification and structural analysis of the CCTO nanocrystals were conducted on a Phillips X’pert Pro X-ray diffractometer equipped with Cu Kα radiation (λ = 1.54056 Å) at a scan rate of 0.5°/s and 2θ from 10° to 80°, operating at 40 kV and 40 mA. Electron micrographs of the CCTO nanocrystals were taken using a transmission electron microscope (TEM, JEOL 100CXII, JEOL-2010) with accelerating voltages of 100 or 200 kV and a scanning electron microscope (SEM, JEOL-6700). X-ray fluorescence spectrometer (XRF, JEOL-3201Z) and X-ray photoelectron spectroscopy (XPS, Axis Ultra DLD) were applied to characterize the elemental composition and the chemical state of CCTO. For the dielectric measurement, both sides of pellets were covered with gold electrodes using a *dc* sputtering technique, and the properties of the composites with different filler loadings were determined by an impedance/gain-phase analyzer (Agilent Hewlett Packard 4194A) at the range of frequency from 100 Hz to 40 MHz. The spectrometer performed the complex impedance *Z** and gain-phase *θ*. *Z** was converted into complex permittivity by
(2)$$\varepsilon^{*}=1/\left(i\omega C_{0}Z^{*}\right)$$
where *ω* is the angular frequency ω=2πf and $C_{0}=\varepsilon_{0}S/d$ is the empty cell capacitance, where *S* is the sample area, *d* is the thickness, and $\varepsilon_{0}$ is the permittivity of free space, $\varepsilon_{0}$ = 8.854 × 10^−12^ F/m.

## 3. Results and Discussion

### 3.1. Identification of the CCTO Powders

The thermogravimetric–differential analysis (TGA–DTG) measure (Figure 1) of the pre-mixed precursor (black powder with an additional 10 times of NaCl in weight ratio) was carried out to present the thermal evolution. The TGA curve showed three steps of the whole process and a slight weight loss between 30 and 350 °C, which was ascribed to the pretreatment of the precursor at 350 °C. A sharp weight loss was observed ranging from 350 to 600 °C. Each step of the precursor decomposition may include complicated reactions, such as the oxidation of organometallics to form CO_2_, species migration and reactions, etc. At the temperature of 600–750 °C, no noticeable weight loss was observed, corresponding to the formation of crystalline CCTO [17,29]. At ~750 °C, the chloride melts started to vaporize, resulting in weight loss in the third step, illustrating that the molten sodium chloride acted as the essential medium and aided the whole reaction.

The XRD patterns of CCTO powders calcinated at different conditions are shown in Figure 2. The calcination temperature impacted the phase formation of CCTO powders. The powder calcinated at 700 °C for 3 h had a small amount of CCTO. Extending the calcination duration to 5 h increased the CCTO phase, which was indicated by intensity enhancement of the apparent (220) peak. Even though it prolonged the reaction time, the impurity phases of CuO, CaTiO_3,_ and TiO_2_ were detected in these two powders, revealing that 700 °C was not high enough to produce a pure CCTO phase. With the increase in calcination temperature to 800 °C, only a minor fraction of CuO and CaTiO_3_ can be detected at 1 h of calcination. The pure phase CCTO can be produced by calcination at 800 °C for 3 h. The XRD patterns confirmed the formation of the CaCu_3_Ti_4_O_12_ phase (JCPDS#75-2188). The peak positions matched quite well with the main diffraction index of body-centered cubic (*bcc*) CCTO with space group *Im3*. Neither peak of discernable titanium oxide, calcium, nor other copper oxide compounds was detected. The lattice parameters were calculated using JADE 6.5 peak fitting and Cohen’s least mean square method, such as *a* = 7.387 Å, comparable to the value reported in the literature [8,30] and JCPDS#75-2188 for 7.391 Å.

Furthermore, the calcination temperature was increased to 900 °C, and plenty of impurity peaks were found in the XRD patterns, which had not been reported in the literature before. These impurity phases were not easy to identify, and they could contain Na_x_Cu_y_O_z_, Ca_x_Cu_y_O_m_Cl_n_, Na_x_Ti_y_O_z_, etc. This phenomenon can refer to the high activity of the vaporized sodium chloride in a free molecular state.

In its pure phase, CCTO can be successfully synthesized at 800 °C for 3 h; the consequential study of the molten salt effect was based on this condition. The morphologies of the precursor, CCTO calcinated with different quantities of sodium chloride, are shown in Figure 3. The samples were labeled as CCTO-800-3 h-0 t, CCTO-800-3 h-3 t, and CCTO-800-3 h-10 t for the additional sodium chloride of 0, 3, and 10 times. Figure 3a shows an amorphous morphology of the black powder precursor, which was confirmed by the XRD patterns in Figure 4. No crystalline peak was detected, which suggested no separated phase of single metal oxide and demonstrated the molecular level mixing of the metal ions. Remarkable differences can be found in Figure 3b–d, where the amount of molten salt played an essential role in the particle size of the CCTO particles, which implied that crystal size decreased with the increase in salt amount. The molten salt diluted the concentration of the nucleus and controlled the growth of the crystals. In Figure 3d, ~70 nm of CCTO nanocrystals were synthesized by the molten salt assisted method, while the grain boundaries were also very clear.

When the XRD patterns of CCTO-800-3 h-0 t and CCTO-800-3 h-10 t were compared, the full width at half maximum (FWHM) of characteristic peaks (220), (400), and (422) became broader as the molten salt amount increased, resulting in smaller particle size according to the Debye–Scherrer equation [17,31] The average crystallite size of CCTO-800-3 h-10 t was calculated to be 56.2 nm according to the four most intense peak inflections at 2*θ* corresponding to 34.29° (220), 49.28° (400), 61.40°(422), and 72.25°(440) after instrument correction. The result was comparable to the observation of TEM.

In ceramic, an internal barrier lay capacitor (IBLC) structure was pronounced during the sintering process to fabricate CCTO ceramic with giant dielectric permittivity. Furthermore, the defect mechanisms (i.e., *n*-type electron or *p*-type hole conduction) had a significant impact on the dielectric permittivity of the final CCTO ceramic. Herein, a post-treatment was applied to enhance the intrinsic permittivity of the CCTO nanocrystal powder.

Figure 5 displays the XRD patterns of CCTO and two samples after 1050 °C post-annealing. Unexpected minor diffraction peaks of CaTiO_3_ were detected in the post-treated samples, exhibiting Cu-deficiency non-stoichiometry. This phenomenon may be due to Cu segregation or volatilization and the formation of Cu vacancies, which coincided with the phase diagram study report in the literature [32,33,34]. According to the XRD pattern, the particles of CCTO increased to 130 nm and 160 nm after 1 h and 4 h treatment, respectively, at 1050 °C. During heat treatment, Cu(II) became unstable and reduced to Cu(I). A slight substitution of Ti(IV) occupied the Cu site to maintain the average charge balance for the compound. On the cooling stage, Cu^+^ was oxidized to Cu^2+^ with released electrons to the Ti *3 d* conduction band, leading Ti^4+^ to Ti^3+^ [17,35,36].

In Figure 6, the peak of Cu 2*p*_2/3_ spectra can be divided into three peaks by Gaussian–Lorentzian profile fitting, which correspond to Cu^+^, Cu^2+^, and Cu^3+^ ions from low to high binding energies. According to the literature, [37,38,39,40], the existence of Cu^+^, Cu^2+^, and Cu^3+^ may be ascribed to the semiconductivity of grains due to the hopping of charge carriers. In addition, this semiconductivity was easy to be detected in the titanate-based materials due to the small but significant amount of oxygen loss at high temperatures (>1000 °C) [23,36,41,42]. The giant dielectric response of CCTO can be clarified by the understanding of the semiconductive nature of the grains. As seen in Figure 6, there was a slight increase in the ratio of Cu^3+^/Cu^2+^ after post-heat treatment, contributing to the enhancement of the dielectric constant according to the previous study [43,44]. These results fitted three models of the defect mechanism, solid solution, and compensation mechanism as follows:(3)$$3{\mathrm{Cu}}^{2+}\to 2{\mathrm{Cu}}^{1+}+{\mathrm{Ti}}_{\mathrm{Cu}}^{4+}+\mathrm{Cu}↑\;\left[\mathrm{Ca}({\mathrm{Cu}}_{3-3v}^{2+}{\mathrm{Cu}}_{2v}^{1+}{\mathrm{Ti}}_{v}^{4+}){\mathrm{Ti}}_{4}O_{12}\right]$$
(4)$${\mathrm{Cu}}^{+}+{\mathrm{Ti}}^{4+}\to {\mathrm{Cu}}^{2+}+{\mathrm{Ti}}^{3+\;}\;\left[\mathrm{Ca}\left({\mathrm{Cu}}_{3-v}^{2+}{\mathrm{Ti}}_{v}^{4+}\right){\mathrm{Ti}}_{4-2v}^{4+}{\mathrm{Ti}}_{2v}^{3+}O_{12}\right]$$
(5)$$3{\mathrm{Cu}}^{2+}\to V_{\mathrm{Cu}}^{\prime \prime }+2{\mathrm{Cu}}^{3+}+\mathrm{Cu}↑\;\left[\mathrm{Ca}({\mathrm{Cu}}_{3-3v}^{2+}{\mathrm{Cu}}_{2v}^{3+}V_{v}^{\prime \prime }){\mathrm{Ti}}_{4}O_{12}\right]$$

### 3.2. Microstructure and Dielectric Properties of CCTO@PVDF Composites

The XRD patterns of CCTO@PVDF composites with 40% volume fraction are shown in Figure 7. For the pure PVDF polymer, several peaks were centered from 15–28°, α/γ (020), α/γ (110), and α (021)/γ(022) at diffraction angles of 2θ = 18.37, 19.94, and 26.46°, respectively, and identified as the characteristic peaks of PVDF. It was confirmed that the PVDF polymer was the mixed phase of α (the most common polymorph) and γ-phase. With the incorporation of CCTO powder in the polymer matrix, the sharp peaks and crystallinity of CCTO revealed that the steady and solid solution was fabricated with the ceramic filler and polymer matrix [45].

Figure 8a displayed the SEM morphology of CCTO-1050-1 h ceramic powder, which depicted the nano-sized particles of 70–200 nm aggregated to some extent after post-heat treatment at 1050 °C. Figure 8b shows the microstructure of neat PVDF, which revealed that the PVDF polymer formed a continuous phase. Figure 8c,d presented the cross-section and surface SEM images of CCTO-1050-1 h@PVDF (40 vol%), illustrating that the CCTO particles were uniformly dispersed in the PVDF matrix. The EDX mapping of the composite presented the distribution of CCTO in the matrix. CCTO nanocrystals surrounded the PVDF network (dark region), forming nano and micro-size “capacitors”.

The dielectric constants (ε) and dielectric loss (tan δ) of CCTO@PVDF composites with various volume fractions of CCTO fillers were measured at room temperature (20 °C), with the frequency dependence ranging from 100 to 40 MHz, and the results were elucidated in Figure 9. From Figure 9a,c, the dielectric constants for two types of CCTO@PVDF composites rose with the increase in CCTO filler loading at all frequencies. The dielectric constants of these composites were higher than that of neat PVDF, as expected but much lower than that of CCTO ceramic [46]. Apart from the compatibility, particle size, and connectivity effect, the non-polar nature of PVDF with constrained polymer chains hindered the contribution of electrical polarization, resulting in the low dielectric constant of the composites.

As shown in Figure 9b,d, the dielectric loss of the CCTO@PVDF composite increased gradually with the CCTO loading. It was observed that the tan δ decreased in the frequency range of 100–10^5^ Hz, but increased substantially over 10^5^ Hz. The increase in dielectric loss at high frequency was related to the glass transition relaxation of PVDF. In contrast, a similar tendency at low frequency can be attributed to the glass transition relaxation associated with molecular motion in the crystalline regions. The dielectric constant of CCTO-800-3 h@PVDF and CCTO-1050-1 h@PVDF showed a distinct difference with the same filler content, especially for the samples with high volume fractions. This result suggests that the filler, the post-treated nano-sized CCTO, was the key to this issue. In our case, the pure phase nano-sized CCTO particles of ~70 nm were successfully synthesized via the two-step molten salt assistance method, and sample CCTO-1050-1 h exhibited a Cu-deficiency property after heat treatment. Cu-deficiency may play a vital role in the giant dielectric constant of CCTO and be fundamental for the formation of insulating barrier layer structures to develop interfacial polarization, but the mechanism is still not precise [47,48]. Although post-treatment resulted in some aggregation of the CCTO nanocrystals, the increase in the grain boundaries may lead to a considerable increase in the intrinsic dielectric constant. In addition, the distribution of CCTO in the polymer matrix was also critical. The nano-sized CCTO can be incorporated into the polymer matrix homogeneously, forming nano-sized “capacitors”, which may significantly improve the dielectric constant of the composite. Furthermore, the interfacial effect greatly enhanced the dielectric properties. With the interfacial area increasing, the electron trapped by Cu^+^ and Ti^3+^ ions and oxygen vacancies caused by Cu^3+^ accumulated at the interface between the filler and matrix and generated the interfacial polarization.

The dielectric constant of the CCTO-800-3 h@PVDF nanocomposite was about 22, with a volume fraction of 7% at 100 Hz. As the concentration further increased to 11%, 26%, 52%, and 64%, the value of ε increased to 26, 39, 110, and 173, respectively, accompanied by the dielectric loss increase. Meanwhile, the CCTO-1050-1 h@PVDF exhibited an apparent elevation on the dielectric constant with 939 at the CCTO filler loading of 64%, and with the value of *ε* as 22, 31, 87, 298, and 582 with concentrations of 7%, 11%, 26%, 40%, and 52%, respectively. The primary reason for this difference was the increase in the intrinsic dielectric constant of nano-sized CCTO post-treatment.

The dielectric constant of CCTO was nearly independent of frequency (up to 10^5^ Hz) over a broader range of temperatures. As seen in Figure 9a,c, the disparities of dielectric constant in 100 Hz to 10^5^ Hz increased in association with the CCTO loading. A sharp decline was observed in the composite at 64 vol% CCTO loading and a sudden increase in dielectric loss appeared. It was apparent that, at the high concentrations of ceramic filler loading (>40 vol%), the nanoparticles established a semiconductivity network which resulted in a dramatic decrease in dielectric constant and a rapid increase in the dielectric loss.

The dielectric properties of the composites at 1 kHz and 1 MHz were summarized in Figure 10. For the CCTO-800-3 h@PVDF composite, the dielectric constant increased from 10 to 54 and the dielectric loss from 0.187 to 0.256 with the increase in CCTO filler content from 0 to 64 vol%, while the dielectric constant increased from 10 to 83 and the dielectric loss from 0.187 to 0.293 with the increase in CCTO filler content from 0 to 52 vol% for the CCTO-1050-1 h@PVDF composite. The composites presented a slight dielectric loss increase and a relatively sharp rise in dielectric constant at 1 MHz, which demonstrated that the composites had great potential in high-frequency usage. Moreover, the curves indicated that 40 vol% might be the critical filler concentration for the homogeneous dispersion. The main reason was that a large amount of high semiconductivity of nano-sized CCTO would improve the dielectric constant of the composite but increase the dielectric loss, which meant that the two dielectric constants needed to be balanced.

To understand the mechanism and predict the dielectric constant of the composites, various models were applied to calculate the theoretical dielectric constant of the composites. For comparison purposes, the experimental dielectric constant of the CCTO-1050-1 h@PVDF composites at 10 kHz with different volume fractions is illustrated in Figure 11.

One of the most common dielectric mixture rules is the Lichtnecker model, referred to as the logarithmic law of mixing. The model can be expressed as follows:(6)$$\ln\varepsilon =\left(1-\emptyset_{f}\right)\ln\varepsilon_{m}+\emptyset_{f}\ln\varepsilon_{f}$$
where *ε_m_, ε_f_,* and *ϕ_f_* are the dielectric constants of PVDF, CCTO, and the volume fraction of the CCTO filler, respectively. The values in this work were substituted for *ε_m,_* and *ε_f_* which are 12 and 10,000 (CCTO-1050-1 h)/5000 (CCTO-800-3 h). In the low concentrations, the *ε* value was comparable to the experimental results, especially for CCTO-1050-1 h@PVDF. At the high concentrations (>40 vol%), the predicted value of *ε* deviated significantly from the experimental data. The Lichtnecker model was generally valid for low filler contents, and deviations from predictions increased with increasing filler contents by the imperfect dispersion of ceramic particles and porosity in the composite at higher filler contents.

The Maxwell–Garnett model describes the dielectric behaviors of the mixture comprising uniform distribution, high dielectric constant spherical ceramic filler, and a matrix of low dielectric constant.
(7)$$\varepsilon =\varepsilon_{m}\frac{2\left(1-\emptyset_{f}\right)\varepsilon_{m\;}+\left(1+2\emptyset_{f}\right)\varepsilon_{f}}{\left(2+\emptyset_{f}\right)\varepsilon_{m}+\left(1-\emptyset_{f}\right)\varepsilon_{f}}=\varepsilon_{m}\left[1+\frac{3\emptyset_{f}\left(\varepsilon_{f}-\varepsilon_{m}\right)}{\left(1-\emptyset_{f}\right)\left(\varepsilon_{f}-\varepsilon_{m}\right)+3\varepsilon_{m}}\right]$$

It can be found that the Maxwell–Garnett model failed to fit the experiment result due to the different properties of these two phases in the mixture.

Jaysundere and Smith proposed a model for predicting the dielectric constant as the randomly connected state in the 0–3 composites and considered the interactions between the adjacent spheres. The equation is given by:(8)$$\varepsilon =\frac{\varepsilon_{m}\left(1-\emptyset_{f}\right)+\varepsilon_{f}\emptyset_{f}\frac{3\varepsilon_{m}}{\left(2\varepsilon_{m}+\varepsilon_{f}\right)}\left[1+3\emptyset_{f}\frac{\left(\varepsilon_{f}-\varepsilon_{m}\right)}{2\varepsilon_{m}+\varepsilon_{f}}\right]}{1-\emptyset_{f}+\emptyset_{f}\frac{3\varepsilon_{m}}{\left(2\varepsilon_{m}+\varepsilon_{f}\right)}\left[1+3\emptyset_{f}\frac{\left(\varepsilon_{f}-\varepsilon_{m}\right)}{2\varepsilon_{m}+\varepsilon_{f}}\right]}$$

As far as the model is concerned, the interaction between the adjacent spheres, which is usually used for high filler content, was deviated from experimental results.

The dielectric constant of a composite is affected by the homogeneity of distribution, the size of the fillers, and the interface between ceramics and polymers. Wong [49] established the effective medium theory model (EMT) to take into account the morphology of the filler, and the equation is:(9)$$\varepsilon =\varepsilon_{m}\left[1+\frac{\emptyset_{f}\left(\varepsilon_{f}-\varepsilon_{m}\right)}{\varepsilon_{m}+n\left(1-\emptyset_{f}\right)\left(\varepsilon_{f}-\varepsilon_{m}\right)}\right]$$
where *n* is the ceramic morphology fitting factor. This model is suitable for ceramic filler sizes of less than 1 μm. The experimental data fitted quite well with the EMT model with the shape parameters *n* = 0.181 (R^2^ = 0.975) and 0.0495 (R^2^ = 0.991) for CCTO-800-3 h@PVDF and CCTO-1050-1 h@PVDF. The good agreements were ascribed to the excellent distribution of the particles. Yamada also proposed a model to predict the dielectric constant of the composite,
(10)$$\varepsilon =\varepsilon_{m}\left[1+\frac{k\emptyset_{f}\left(\varepsilon_{f}-\varepsilon_{m}\right)}{k\varepsilon_{m}+\left(1-\emptyset_{f}\right)\left(\varepsilon_{f}-\varepsilon_{m}\right)}\right]$$
where *k* is the geometry of the ceramic particle, the geometry affecting the connectivity and electricity, further affecting the dielectric constant. Based on the two models, we can deduce that when the EMT model and Yamada model are expressed in the same equation while *n* = 1/*k,* then the corresponding *k* for the two composites are 5.525 and 20.20, respectively.

The dielectric constant of the composite can be improved by introducing the high dielectric constant ceramic filler, but the ceramic filler is different from its sintered form. Moreover, the mechanism of the dielectric properties for the ceramic polymer is complex, and it is not only dependent on the dielectric constant of the matrix and filler, size, particle distribution, and filler concentration, but also on the interphase between the filler and matrix. The only difference between the composites is the different fillers. The CCTO-1050-1 h with a Cu-deficiency property enhanced interfacial polarization, and the oxygen vacancies at grain boundaries transferred to the matrix at the interfacial region. In addition to the small size and large interfacial area, the dielectric constant of the composite is higher than in other research [20]. However, the dielectric loss of CCTO-1050-1 h@PVDF can be reduced by introducing the coupling agent to moderate the qualitative difference between the inorganic and organic phases [50].

## 4. Conclusions

In this study, ~70 nm of CCTO nanocrystals were successfully synthesized by precursor and calcinated at 800 °C for 3 h with the assistance of 10 times molten salt. The CCTO nanocrystals, after post-treatment at 1050 °C, obtained Cu-deficiency CCTO nanocrystals. The ceramic–polymer composite was fabricated based on these two CCTOs. The SEM and mapping results show that the fillers were homogeneously incorporated into the PVDF matrix. The dielectric properties of the composites were measured, ranging from 100 Hz to 40 MHz in frequency and at room temperature. The results depict the dielectric constant and loss increased with the increasing volume fraction in the CCTO nanoparticles. When the filler content was of 64 vol%, the dielectric constant of the composite reached as high as 939, which is 78 times higher than the pure PVDF with a moderately increased dielectric loss at 100 Hz, and the composites showed potential in the high frequency (10^6^ Hz) application. Several theoretical models were adopted to fit the experimental data and predict the dielectric constant of the composites at various volume fractions of the filler, among which the EMT and Yamada models were in good agreement with the experimental results. The dielectric properties characterization data implied that the post-treated CCTOs could fabricate the high dielectric constant composites.

## Figures and Tables

**Figure 1 polymers-14-04328-f001:**
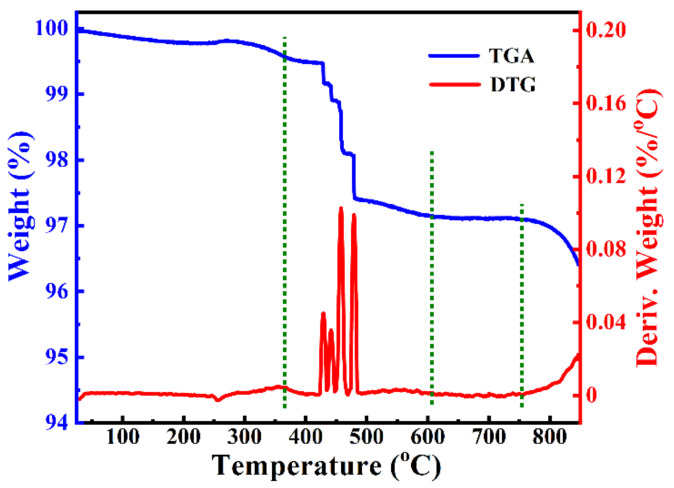
Thermogravimetric–differential analysis (TGA–DTG) measurement of pre-mixed precursor (black powder with additional 10 times NaCl in weight ratio) at a heating rate of 10 °C/min in air atmosphere.

**Figure 2 polymers-14-04328-f002:**
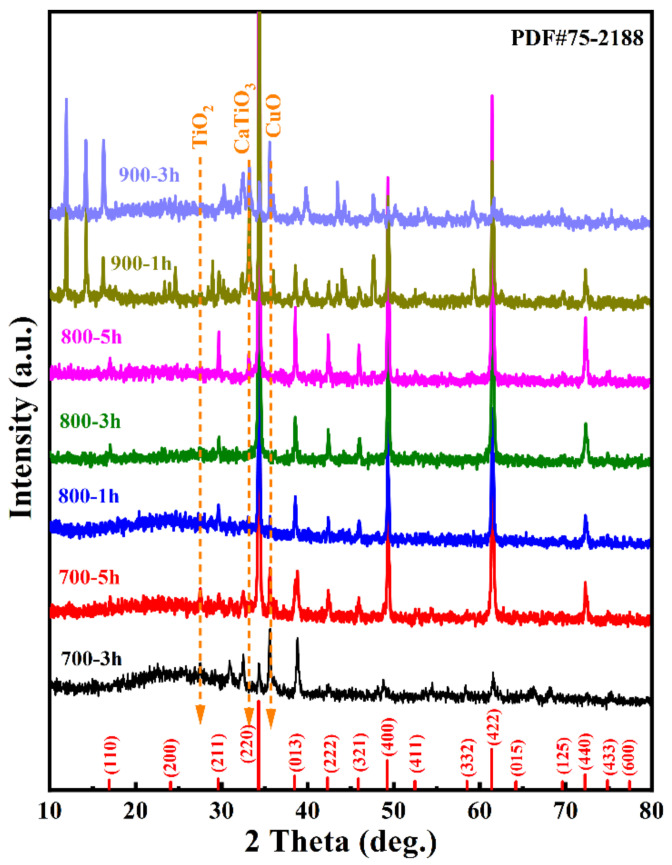
XRD patterns of CCTO powders calcinated at different conditions. Red vertical lines represent the standard pattern of CCTO (JCPDS #75-2188).

**Figure 3 polymers-14-04328-f003:**
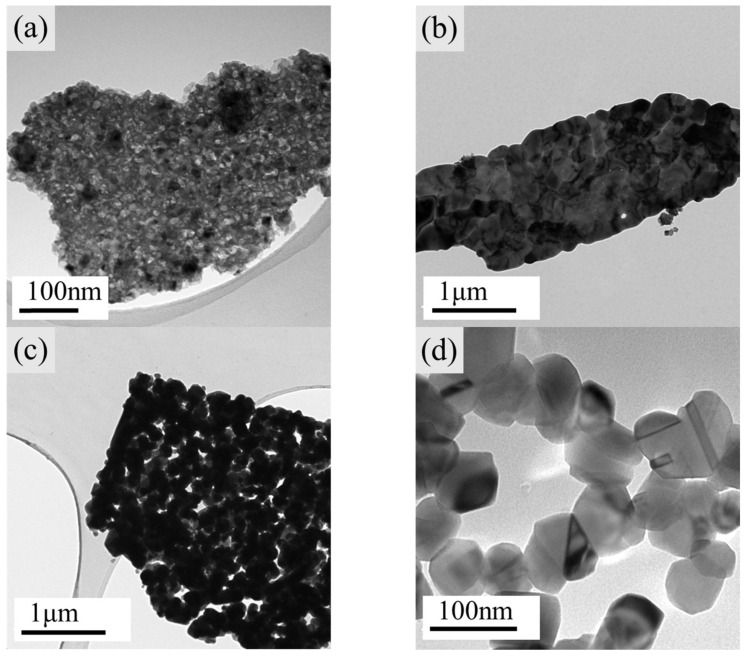
TEM images of (**a**) black powder precursor, (**b**) CCTO-800-3 h-0 t, (**c**) CCTO-800-3 h-3 t, and (**d**) CCTO-3 h-10 t.

**Figure 4 polymers-14-04328-f004:**
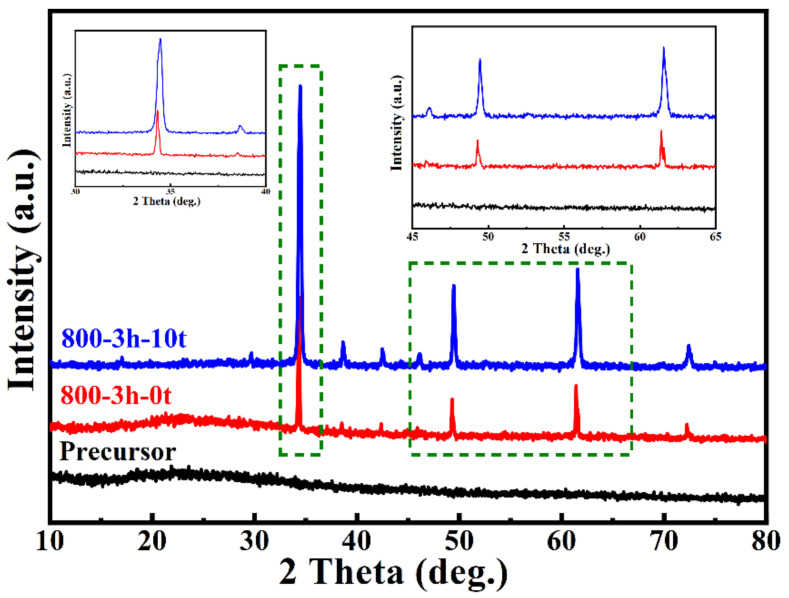
XRD patterns of the black precursor powder, CCTO-800-3 h-0 t, and CCTO-800-3 h-10 t; and the enlarged spectra of selected rectangle areas (insert figures).

**Figure 5 polymers-14-04328-f005:**
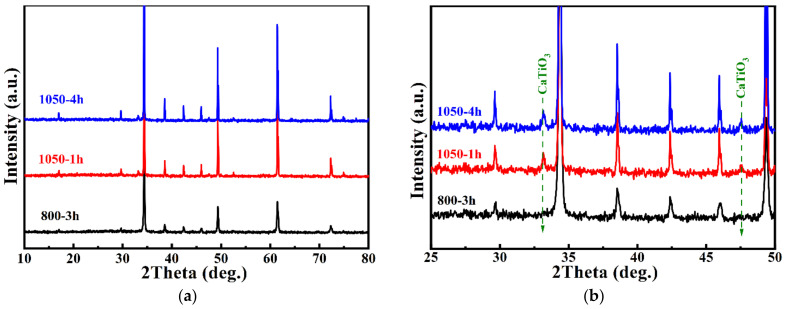
(**a**) XRD patterns of CCTO-800-3 h, CCTO-1050-1 h, and CCTO-1050-4 h; and (**b**) the enlarged spectra of (**a**).

**Figure 6 polymers-14-04328-f006:**
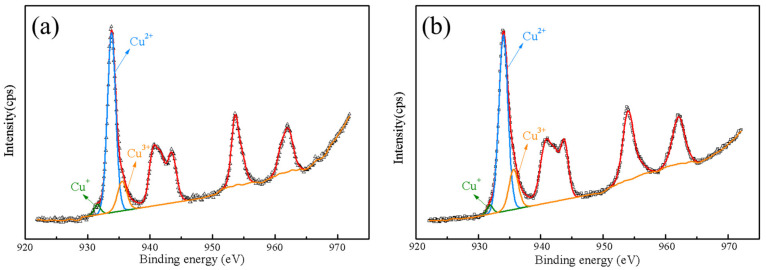
Cu 2*p*_3/2_ XPS spectra of (**a**) CCTO-800-3 h and (**b**) CCTO-1050-1 h nanocrystals.

**Figure 7 polymers-14-04328-f007:**
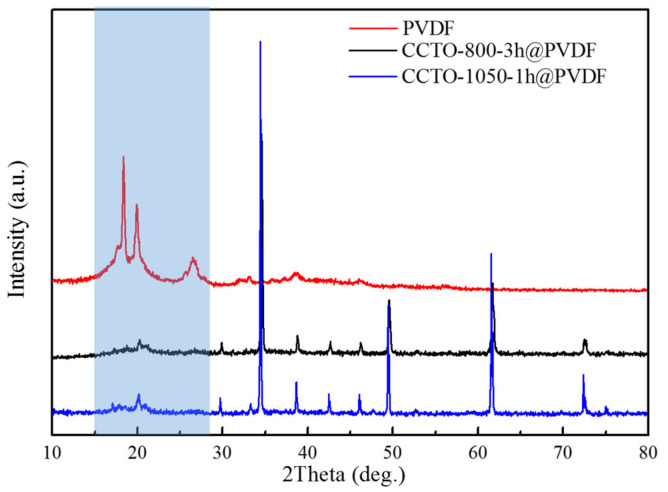
XRD patterns of CCTO@PVDF composites with different CCTO (40 vol%).

**Figure 8 polymers-14-04328-f008:**
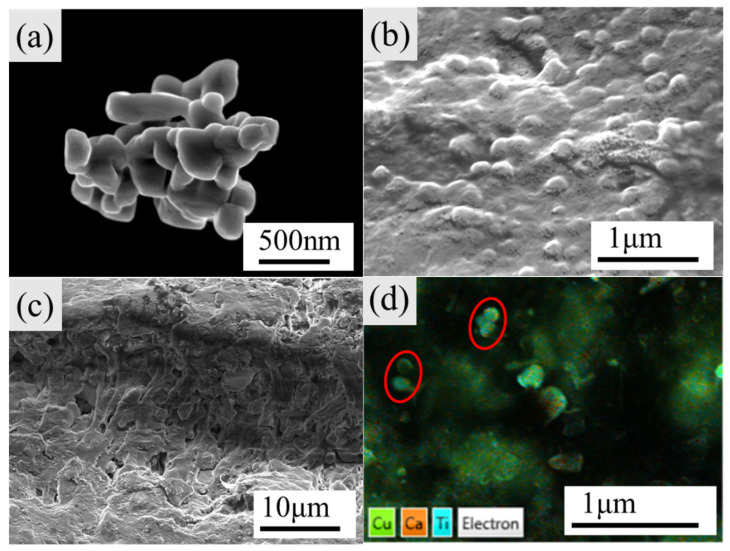
SEM images of sample (**a**) CCTO-1050-1 h, (**b**) surface of neat PVDF, (**c**) cross-section of CCTO-1050-1 h@PVDF composite (40 vol%), and (**d**) EDX mapping of surface of CCTO-1050-1 h@PVDF composite (40 vol%), the red circles present the CCTO particles.

**Figure 9 polymers-14-04328-f009:**
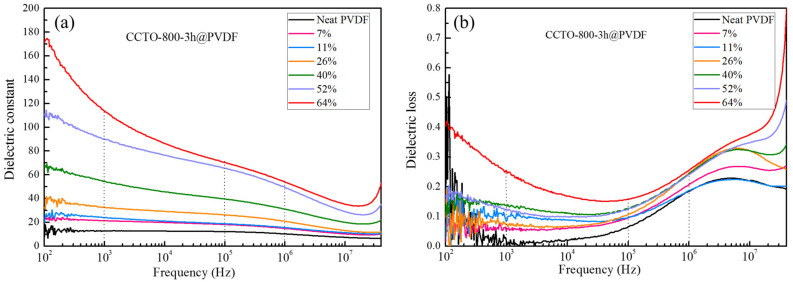
The frequency dependence of dielectric constants and (ε) and dielectric loss (tan δ) of CCTO-800-3 h@PVDF (**a**,**b**) and CCTO-1050-1 h@PVDF (**c**,**d**) at various volume fractions at room temperature.

**Figure 10 polymers-14-04328-f010:**
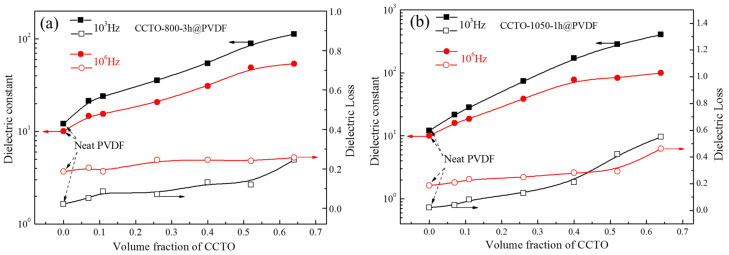
Dielectric properties at 1 kHz and 1 MHz of composites (**a**) CCTO-800-3 h@PVDF and (**b**) CCTO1050-1 h @PVDF at different filler contents.

**Figure 11 polymers-14-04328-f011:**
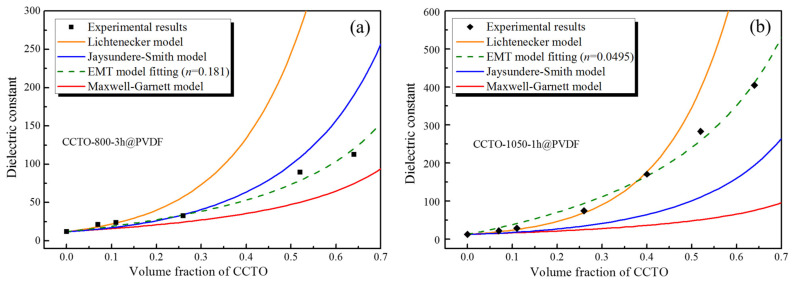
Various theoretical predictions and experimental results of the dielectric constant of (**a**) CCTO-800-3 h@PVDF and (**b**) CCTO-1050-1 h@PVDF composites at different volume fractions of filler at 1 kHz.

## Data Availability

The data presented in this study are available upon request from the corresponding author. The data are not publicly available due to the laboratory management regulations.

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
