# Peer review of "Nano-Sized Calcium Copper Titanate for the Fabrication of High Dielectric Constant Functional Ceramic–Polymer Composites"

_polymers, 2022, doi:10.3390/polym14204328_

Round 1
Reviewer 1 Report
The authors studied " Nano sized CCTO for the fabrication of high dielectric constant functional ceramic polymer composite.
There are many research works has been carried on this, and hence the following observations:
1. The authors used very lengthy procedure for the fabrication of nano CCTO ceramics. What is the novelty in this?
2. There are reports on PVDF+CCTO composites fabricated using both nano and micro CCTO has been published in the literature. Authors should add these references and discuss these in the introduction part.
3. What is the dielectric constant value obtained for the copper deficient CCTO ceramic.
4. How many measurements were made for calculating the dielectric constant of the composites and what is the error levels?
5. What is the composite connectivity for the compression molded polymer -ceramic composite?
Author Response
Response to Reviewers
20-Sep-2022
Dear Reviewers,
Thank you very much for your constructive comments and suggestions concerning our manuscript entitled “Nano-sized CCTO for the fabrication of high dielectric constant functional ceramic polymer composite” (Manuscript ID: polymers-1919386). The comments are helpful for us to improve the quality of our work. We have carefully studied your comments and substantially revised our manuscript, hoping to meet the high standard for publication on Polymers. The revisions are addressed point by point below, and the changes are highlighted in the revised manuscript.
To Reviewer #1:
Comment 1: The authors used very lengthy procedure for the fabrication of nano CCTO ceramics. What is the novelty in this?
Reply: Thank you for your comment. The traditional method to prepare CCTO is the sol-gel method. Usually, the CCTO particles synthesized by the sol-gel method reach the micron level, which is not conducive to uniform dispersion in the PVDF polymer matrix. In our work, the CCTO nanocrystals were prepared using the molten salt-assisted method, and the dielectric constant of the CCTO@PVDF nanocomposite is higher than in other research. The EMT and Yamada models are well-agreed with the experimental results, indicating excellent polymer matrix distribution and enhancing the dielectric constant. However, the mechanism of dielectric properties for the ceramic-polymer is complex, which is not only dependent on the dielectric constant of the matrix and filler, size, particle distribution, and filler concentration, but also on the interphase between the filler and matrix. The CCTO-1050-1h with Cu-deficiency property enhanced interfacial polarization, and the oxygen vacancies at grain boundaries would transfer to the matrix at the interfacial region. Please Check Line 352 on Page 11.
Comment 2: There are reports on PVDF + CCTO composites fabricated using both nano and micro CCTO have been published in the literature. Authors should add these references and discuss these in the introduction part.
Reply: Thank you for your kind reminder. We have provided the novelty of the work at the end of the introduction section, as shown below. Please check the additional content on Page 2 in the main text.
Thank you for your suggestion. There have been some papers on the formation of PVDF+CCTO composites at the micron level by the sol-gel method, and these papers have been discussed in the introduction part. “Chao et al. reported CCTO/PVDF nanocomposites (50 vol%) with a high dielectric property of about 62.3 and a low loss of 0.081 using a series of synthesis methods.[21] Hu et al. studied a unique polymer composite fabricated with CCTO nanofibers (20 vol%) and PVDF via the electrospinning method. Liu’s research revealed that CCTO nanowire/epoxy efficiently improved the dielectric permittivity and mechanical properties of ceramic-based polymer composites.[22, 23] Wang et al. reported that the dielectric constant of CCTO/PVDF composite materials with 50% CCTO achieved a maximum value of 50 almost, which is 5 times higher than the pure PVDF.[24]” Please have a check in Page 2.
- Chao, XL; Wu, P; Zhao, Y; Liang, PF, and Yang, ZP. Effect of CaCu3Ti4O12 Powders Prepared by the Different Synthetic Methods on Dielectric Properties of CaCu3Ti4O12/Polyvinylidene Fluoride Composites. J. Mater. Sci. - Mater. Electron. 2015; 26 :3044. [CrossRef]
- Hu, CH; Zha, JW; Yang, Y; Zheng, MS, and Dang, ZM. Enhanced Dielectric Properties of Polyvinylidene Fluoride Nanocomposites via Calcium Copper Titanate Nanofibers. 2017 1st International Conference on Electrical Materials and Power Equipment (Icempe) 2017; :258. [CrossRef]
- Liu, YP; Li, LY; Guo, MJ; Zhou, Z; Chen, GX, and Li, QF. Improving Dielectric and Mechanical Properties of CaCu3Ti4O12 Nanowire/Epoxy Composites through a Surface-Polymerized Hyperbranched Macromolecule. ACS Applied Electronic Materials 2019; 1 :346. [CrossRef]
- Wang, JJ; Deng, QJ; He, YY; Feng, YN; Kang, MP; Duan, XL, and Yang, YL. Fabrications and dielectric performances of novel composites: Calcium copper titanate / Polyvinylidene fluoride. Curr. Appl. Phys. 2022; 39 :25. [CrossRef]
Comment 3: What is the dielectric constant value obtained for the copper deficient CCTO ceramic.
Reply: Thank you for your comment. We tried to fabricate the Cu-CCTO ceramic pellet and measure the dielectric constant; however, due to the immense strength of the alligator clip, the pellet clasped and failed in the test. And, we believe that the adhesive additive and sintered target would affect the inherent dielectric constant, so we only obtained the dielectric constant of ceramic-polymer nanocomposites.
Comment 4: How many measurements were made for calculating the dielectric constant of the composites and what is the error levels?
Reply: Thank you for your comment. We calculate the dielectric constant of our composites, typically performing 5 measurements with an error level of less than 6%.
Comment 5: What is the composite connectivity for the compression molded polymer -ceramic composite?
Reply: Thank you for your comment. According to related references[1,2], the composite connectivity for the compression molded polymer ceramic composite is 0−3 type.
- Cao, M; Li, L; Hong, WB; Wu, SY, and Chen, XM. Greatly enhanced permittivity in BaTiO3-epoxy dielectric composites with improved connectivity of ceramic phase. Journal of Materiomics 2021; 7 :1. [CrossRef]
- Cao, M; Yan, XJ; Li, L; Wu, SY, and Chen, XM. Obtaining Greatly Improved Dielectric Constant in BaTiO3-Epoxy Composites with Low Ceramic Volume Fraction by Enhancing the Connectivity of Ceramic Phase. ACS Appl. Mater. Interfaces 2022; 14 :7039. [CrossRef]
Thank you again for the critical comments and helpful suggestions.
Sincerely yours,
Dr. Shaojuan Crystal LUO (on behalf of all co-authors)

Reviewer 2 Report
The paper presents a study about calcium copper titanate in nano scale in order to receive high dielectric constant functional ceramic polymer composites.
Authors prepared and investigated novel calcium copper titanate (CaCu3Ti4O12)-polyvinylidene fluoride composite (CCTO@PVDF) with Cu-deficiency. They characterized the morphology and structure of polymer composites uniformly incorporated with CCTO nanocrystals. Authors obtained relatively high dielectric constant of 939 was obtained at 64% vol% CCTO@PVDF content, 78 times that of pure PVDF. Mentioned results showed that it was successfully proposed that the post-treated strategy greatly enhanced the dielectric properties of ceramic polymer composites.
Thank you very much for the paper about materials characterized by high dielectric constant, added nano particles. There are many papers, focused on this topic, what means that this study is very important. I think, mentioned materials can find applications in many parts of industry, especially in electric power devices, such as insulating materials or others. I put some comments and suggestions.
1. Authors used name “dielectric constant”, what is common name, used by many researchers. The true is that correct name is “relative electrical permittivity”. So, please add in Introduction chapter some explanation that you mean and investigate “electrical permittivity”.
2. The topic of the paper is “dielectric constant”. So, I would expect some fundamental information about the constant. What does it mean. Some equation, some physical sense of the constant. Please complete the Introduction chapter.
3. Chapter Materials well describes used materials and methods. Anyway, I did not find any explanation why authors used exactly mentioned materials. Also, why did authors used exactly mentioned % of nano particles. Please explain it.
4. Authors studied dielectric losses – tan(delta). I would expect also some explanation what this parameter means, and what it describes. Please complete.
5. Fig. 9 presents epsilon and tan(delta) as a function of frequency. Please describe why the characteristics look like.
Author Response
Response to Reviewers
22-Sep-2022
Dear Reviewers,
Thank you very much for your constructive comments and suggestions concerning our manuscript entitled “Nano-sized CCTO for the fabrication of high dielectric constant functional ceramic polymer composite” (Manuscript ID: polymers-1919386). The comments are helpful for us to improve the quality of our work. We have carefully studied your comments and substantially revised our manuscript, hoping to meet the high standard for publication on Polymers. The revisions are addressed point by point below, and the changes are highlighted in the revised manuscript.
To Reviewer #2:
Comment 1: Authors used name “dielectric constant”, what is common name, used by many researchers. The true is that correct name is “relative electrical permittivity”. So, please add in Introduction chapter some explanation that you mean and investigate “electrical permittivity”.
Reply: Thank you for your kind reminder. Indeed, the dielectric constant, also known as the relative electrical permittivity, is a physical quantity that expresses the properties of a dielectric and is equal to the ratio of the capacitance value C of a capacitor when it is filled with a uniform dielectric to the capacitance value when it is a vacuum. Generally speaking, the higher the dielectric constant, the greater the capacitance value of its capacitor.
We have added this section to the introduction; Please check Lines 40 on Page 1.
Comment 2: The topic of the paper is “dielectric constant”. So, I would expect some fundamental information about the constant. What does it mean. Some equation, some physical sense of the constant. Please complete the Introduction chapter.
Reply: Thank you for your kind suggestion. Thank you for your comment. For a polymer dielectric capacitor, the energy density Ue is proportional to its dielectric constant K and the square of the electric field E:
Where É›0 is the vacuum permittivity Dielectric constant is a dimensionless factor, usually expressed as É›, which is always relative to the vacuum dielectric constant (É›0 = 8.854×10‒12 F m-1).[9] Please check Line 40-45 on Page 1 and Line 118 on Page 3.
- Xie, X; Yang, C; Qi, X-d; Yang, J-h; Zhou, Z-w, and Wang, Y. Constructing polymeric interlayer with dual effects toward high dielectric constant and low dielectric loss. Chem. Eng. J. 2019; 366 :378. [CrossRef]
Comment 3: Chapter Materials well describes used materials and methods. Anyway, I did not find any explanation why authors used exactly mentioned materials. Also, why did authors used exactly mentioned % of nano particles. Please explain it.
Reply: Thank you very much for the suggestions. “There are some ferroelectric ceramic materials such as Pb(Zr,Ti)O3, BaTiO3, Pb(Mg1/3Nb2/3)O3-PbTiO possess high dielectric constant. In the theoretical study, CCTO ceramic has a giant dielectric constant and remains unchanged in a wide temperature range (100-500K).” As a result, we used the CCTO ceramic material as the filler. The density of the PVDF and as-prepared CCTO is 1.8g/cm3 and 3.2g/cm3, respectively. The mentioned vol% of nano particles is calculated from the mass proportion. Please check Line 48 on Page 2.
Comment 4: Authors studied dielectric losses – tan(delta). I would expect also some explanation what this parameter means, and what it describes. Please complete.
Reply: Thank you for your good suggestion. Dielectric loss is the dielectric in the electric field; in the unit of time due to heat and consumption of energy, also known as dielectric loss power, dielectric loss is applied to the AC electric field in the dielectric one of the important quality indicators. The dielectric loss not only consumes electrical energy but also makes the component heat affect its regular work. If the dielectric loss is significant and even causes dielectric overheating and insulation damage, so the smaller the dielectric loss, the better.
Comment 5: Fig. 9 presents epsilon and tan(delta) as a function of frequency. Please describe why the characteristics look like.
Reply: Thank you for the comment. Some polarization mainly causes material dielectric constant formation, polarization is some dipole directional arrangement, due to the change of frequency, the dipole with external field inversion, when the frequency is high, due to the material internal resistance, the dipole reversal to keep up with the speed of the electric field, will form a gallop, medium material is also one of the causes of loss, high-frequency cases, some dipole stop reversal, so the contribution to the dielectric constant is zero. There are generally several polarization methods in materials, and the frequency band of various polarization rotations is different. In general, with the increase of frequency, the dielectric constant generally decreases.
Thank you again for the critical comments and helpful suggestions.
Sincerely yours,
Dr. Shaojuan Crystal LUO (on behalf of all co-authors)
